# Inhibition of Macrophage-Specific CHIT1 as an Approach to Treat Airway Remodeling in Severe Asthma

**DOI:** 10.3390/ijms24054719

**Published:** 2023-03-01

**Authors:** Piotr Sklepkiewicz, Barbara Dymek, Michal Mlacki, Agnieszka Zagozdzon, Magdalena Salamon, Anna Maria Siwińska, Marcin Piotr Mazurkiewicz, Natalia de Souza Xavier Costa, Marzena Mazur, Thais Mauad, Adam Gołębiowski, Karolina Dzwonek, Jakub Gołąb, Zbigniew Zasłona

**Affiliations:** 1Molecure SA, 02-089 Warsaw, Poland; 2Department of Pathology, Faculty of Medicine, University of São Paulo, Avenida Dr. Arnaldo, 455, Room 1150, Cerqueira Cesar, São Paulo 01246-903, Brazil; 3Department of Immunology, Medical University of Warsaw, 02-097 Warsaw, Poland

**Keywords:** macrophage, chitotriosidase, asthma

## Abstract

Chitotriosidase (CHIT1) is an enzyme produced by macrophages that regulates their differentiation and polarization. Lung macrophages have been implicated in asthma development; therefore, we asked whether pharmacological inhibition of macrophage-specific CHIT1 would have beneficial effects in asthma, as it has been shown previously in other lung disorders. CHIT1 expression was evaluated in the lung tissues of deceased individuals with severe, uncontrolled, steroid-naïve asthma. OATD-01, a chitinase inhibitor, was tested in a 7-week-long house dust mite (HDM) murine model of chronic asthma characterized by accumulation of CHIT1-expressing macrophages. CHIT1 is a dominant chitinase activated in fibrotic areas of the lungs of individuals with fatal asthma. OATD-01 given in a therapeutic treatment regimen inhibited both inflammatory and airway remodeling features of asthma in the HDM model. These changes were accompanied by a significant and dose-dependent decrease in chitinolytic activity in BAL fluid and plasma, confirming in vivo target engagement. Both IL-13 expression and TGFβ1 levels in BAL fluid were decreased and a significant reduction in subepithelial airway fibrosis and airway wall thickness was observed. These results suggest that pharmacological chitinase inhibition offers protection against the development of fibrotic airway remodeling in severe asthma.

## 1. Introduction

Asthma is an umbrella term compromising multiple endotypes ranging from easy to control, mild eosinophilic inflammation to severe, steroid-resistant neutrophilic asthma with airway remodeling and persistent air-flow limitation [1,2]. Airway remodeling involves long-term disruption and modification of airway architecture, which significantly contributes to airway hyperresponsiveness (AHR) and lung function decline, which in some cases results in death [3,4]. Post-mortem studies of asthmatic patients reveal that airway remodeling can affect both large and small airways [3]. Due to limited invasive assessments of airway remodeling [5], our understanding of the small airway changes that are relevant to airway remodeling and ultimately to asthma severity is quite poor [6]. Moreover, current asthma treatments involve unspecific steroids to reduce airway inflammation and bronchodilators for exacerbations, but apart from bronchial thermoplasty, none are targeted towards airway remodeling making it an unmet clinical need. Therefore, therapies reversing established remodeling in addition to an anti-inflammatory approach have the potential to improve severe asthma [7].

Macrophages are long-lived immune cells capable of shaping their environment, which makes them an attractive therapeutic target. This is achieved by their high functional plasticity and their pleiotropic role in orchestrating immune responses [8,9]. Although chitotriosidase (CHIT1) can be expressed by various myeloid and epithelial cells, its high expression and secretion is predominantly restricted to activated macrophages [10]. CHIT1 is an enzyme which plays numerous roles in macrophage biology [11] and its increased activity and expression has been implicated in various inflammatory disorders [12,13,14], including lung diseases [15,16,17,18,19]. Lung-resident macrophages are the first responders to environmental assaults and their inhibitory role in the development of allergic airway inflammation has been well documented [20,21]. However, recruitment of monocyte-derived macrophages—a process in which CHIT1 was implicated [22,23]—plays a pathogenic role in allergic airway inflammation [21,24]. Moreover, CHIT1 participates in macrophage polarization [12,18,22,25], a process which is detrimental to asthma [26,27]. Therefore, we hypothesized that inhibition of CHIT1 can alter macrophage biology and have a positive outcome on asthma development. As profibrotic and pro-homeostatic macrophage phenotypes remain ill-defined [28], CHIT1 can also be a useful functional marker for different asthma endotypes.

In the current study we have outlined the beneficial role of pharmacological inhibition of CHIT1 with OATD-01—a previously described chitinase inhibitor [29,30]. OATD-01 demonstrated efficacy in a chronic HDM-induced lung inflammation model that cumulates the detrimental effects of persistent inflammation and airway remodeling. Recent study has shown that nintedanib—a drug approved for lung fibrosis—works by reprograming the profibrotic macrophage phenotype [28]. Here, we present reprograming of profibrotic, CHIT1-expressing macrophages as a successful therapeutic approach to manage airway remodeling in severe asthma. Similar to how checkpoint inhibitors have led to breakthroughs in various solid tumor treatments, a new approach targeting macrophage–fibroblast crosstalk is needed to treat airway remodeling in severe asthma and other interstitial lung diseases (ILDs) [15,29,31,32,33].

## 2. Results

### 2.1. Increased CHIT1 Expression in Fatal Asthma

Comparison of lung specimens obtained from control donors (n = 9) and from individuals with fatal, steroid-naïve asthma (n = 12) revealed increased CHIT1-positive staining localized in the remodeled small airways (Figure 1A–C). Importantly, similar to our previous observations in idiopathic pulmonary fibrosis (IPF) patients, AMCase expression has not been induced in diseased patients. Taken together, our data present CHIT1 as a dominant chitinase activated in fibrotic areas of the lungs of severe asthmatics and suggests that CHIT1 can play a crucial role in fibrotic remodeling of different origins.

### 2.2. Activation of Macrophage-Specific CHIT1 in Chronic House Dust Mite (HDM) Asthma Model

In order to establish a model which can reflect fibrotic changes observed in fatal asthma, we set-up chronic HDM-induced murine models characterized by the presence of inflammatory cells and histopathological evidence of airway remodeling (Figure 2A–E). Chronic administration of HDM significantly induced CD45-positive cells infiltration in the lungs which was highest after 7 weeks of HDM administration (Figure 2A). Due to macrophage-specific localization of CHIT1 in remodeled airways of fatal asthma patients, we chose a 7-week-long HDM administration when alveolar macrophage accumulation (CD11c^pos^ and SiglecF^pos^) was significantly induced (Figure 2B,C). Expression of GR.1-positive cells, reflecting neutrophils and inflammatory monocytes, was below 5% and did not differ among different groups in tested models. Moreover, chitinolytic activity in both BAL fluid (Figure 2D) and plasma (Figure 2E), reflecting local and systemic inflammation, respectively, was increased. The chosen 7-week HDM model was characterized by innate immune activation and specifically inflammatory macrophages, as well as subepithelial and peribronchial fibrosis development and airway remodeling. CHIT1 was specifically expressed by macrophages that were localized near remodeled airways, as reflected by picrosirius red (PSR)–fast green (FG) staining of collagen deposition around the airways (Figure 2F(a–h)). Moreover, chronic inflammation features correlated with CHIT1-positive profibrotic macrophage accumulation.

### 2.3. Chitinase Inhibition with OATD-01 Dose-Dependently Ameliorates Pulmonary Inflammation and Goblet Cell Metaplasia in Chronic Asthma Model

Having established a model where chronic inflammation led to profound airway remodeling, we decided to treat mice with CHIT1 inhibitor (OATD-01) in a therapeutic regimen (Figure 3A). Oral administration of OATD-01 at the previously established dose of 30 mg/kg qd [29] starting from week 5 till 7 exhibited anti-inflammatory properties, as demonstrated by a significant dose-dependent decrease in CD45-positive leukocytes in the lungs as compared to vehicle-treated mice (group termed “Chronic HDM”; Figure 3B). Decreased eosinophil numbers were accountable for the dose-dependent decrease in total cell numbers (Figure 3B). Furthermore, periodic acid–Schiff (PAS) staining and histological analysis revealed a dose-dependent reduction in goblet cells metaplasia after OATD-01 treatment (Figure 3C–D).

### 2.4. OATD-01 Changes Phenotype, but Not Number of Profibrotic Lung Macrophages

Next, we aimed at confirming in vivo target engagement in the lungs and determining the pharmacodynamic profile of OATD-01 in mice subjected to HDM (Figure 4A). We observed a dose-dependent decrease in chitinolytic activity in lung tissue homogenates and plasma after OATD-01 treatment (Figure 4B). This experiment revealed that systemic, but also lung distribution and concentration of, OATD-01 was optimal for CHIT1 inhibition. Goblet cell metaplasia is driven by IL-13 [34,35] and OATD-01 resulted in a dose-dependent reduction in Il13 expression in the lungs (Figure 5A). Importantly, levels of a pro-fibrotic biomarker—an active form of TGFβ1—in BAL fluid (Figure 5B) were also decreased, demonstrating that OATD-01 is an inhibitor of profibrotic pathways, which play crucial role in the development of airway remodeling [36]. Importantly, while total lung macrophage numbers were not significantly changed after OATD-01 treatment (Figure 5C), macrophage-dependent CHIT1 release was (Figure 5D), as evidenced by Western blot analysis of secreted CHIT1 in BAL fluid (Figure 5E). These results revealed that CHIT1 inhibition changed the profibrotic macrophage phenotype rather than their numbers. Our data indicate the potential of OATD-01 in macrophage reprograming and provide an explanation for observed in vivo TGFβ1 inhibition.

### 2.5. Chitinase Inhibition Prevents Development of Airway Remodeling after Chronic HDM Administration

Having established the importance of CHIT1 inhibition in the attenuation of inflammation and goblet cell metaplasia, we decided to specifically focus on the role of OATD-01 in HDM-induced airway remodeling. Quantitative evaluation of bronchial collagen wall and epithelial thickness was performed, demonstrating the inhibition of airway remodeling by OATD-01 (Figure 6A–C). Taken together, OATD-01 significantly reduced subepithelial fibrosis as shown by quantitative analysis of collagen area around the airways (Figure 6B), as well as quantitative analysis of airway wall thickness (Figure 6C).

In summary, we demonstrate that inhibition of macrophage-specific CHIT1 expression changes the phenotype of these cells and results in inhibition of profibrotic TGFβ1 production, and subsequently Th2-dependent IL-13 production. These changes restrain fibroblast activation by inhibition of macrophage–fibroblast positive feedback loop, leading to myofibroblast driven airway remodeling. Overall, the effect results in less collagen deposition in extracellular space and attenuated asthma airway remodeling, which is a fatal feature of this lung disorder (Figure 7).

## 3. Discussion

In the current study we demonstrate for the first time that macrophage-specific CHIT1, but not AMCase, is significantly localized to remodeled airways in fatal steroid-naïve asthmatics. This specific population provided valuable information about significant changes in CHIT1 expression in the lungs of unmedicated severe asthma patients. We present significant antifibrotic properties of the chitinase inhibitor OATD-01 in a chronic airway remodeling HDM model. Inhibition of CHIT1-dependent fibrotic changes in the airways has implications for other lung disorders where profibrotic inflammatory macrophages are progressing disease. We propose that macrophage–fibroblast communication within a lung tissue is altered by macrophage-specific CHIT1 activation. This results in fibroblast to myofibroblast transition, activation, and uncontrolled proliferation leading to airway remodeling. OATD-01 via CHIT1 inhibition can reprogram macrophage profibrotic phenotype and restore macrophage–fibroblast homeostatic interactions which reverse airway remodeling.

Numerous reports show a significant increase in chitinolytic activity in the lungs of patients suffering from chronic lung diseases such as severe asthma [37] or chronic obstructive pulmonary disease (COPD) [19,38,39], but also progressive fibrotic ILDs [15,40] such as IPF [33] and sarcoidosis [29,41], all of which develop lung remodeling. We present CHIT1—the main chitinase expressed in human lung [33,42]—as an enzyme produced by activated macrophages [11] which promotes TGFβ1- and IL-13-driven fibrotic responses [43,44]. CHIT1 was shown to be among the topmost abundantly represented genes of novel profibrotic macrophage populations exclusively present in lungs of patients with lung fibrosis [45]. A recent study by our group [33] described significant macrophage-derived activation of CHIT1 in IPF patients and a pulmonary fibrosis animal model. Similar to data presented here, treatment with the chitinase inhibitor OATD-01 showed significant anti-inflammatory and antifibrotic effects in bleomycin-induced pulmonary fibrosis [33]. These findings further support the idea that profibrotic macrophage accumulation provides a specific source of chitinolytic activity at sites of lung remodeling of various origins and broadens the spectrum of indications for OATD-01. CHIT1 is therefore a profibrotic macrophage functional biomarker contributing to fibrotic lesion progression leading to airway remodeling in severe asthma. Further studies need to be performed to assess whether CHIT1 expression can help to stratify patients with different endotypes of asthma, as it was suggested for COPD patients [31,46].

Initially chitinases were hypothesized to play a significant role in immune response modulation in asthma, owing to the observation that predominant aeroallergens are chitin-containing pathogen particles [47]. However, recent data coming from several reports demonstrate their critical role in facilitating immune response and tissue repair processes in a chitin-independent manner [48]. Specifically, chitinases actively participate in IL-13-driven development of the Type 2 immune response in asthma [48] or TGF-β-driven tissue repair in pulmonary fibrosis [44] in animal models induced by non-chitin-containing insults, namely ovalbumin and bleomycin, respectively. In our study OATD-01 inhibited fibrotic pathways activated by both of these cytokines. Although the breadth of preclinical data determines CHIT1 macrophage-specific expression as a valid therapeutic target, still more research must be conducted to understand non-chitin triggers at sites of injury leading to chronic, uncontrolled inflammation and subsequently airway remodeling.

A 7-week-long HDM model allowed us to evaluate anti-inflammatory and antifibrotic features of chitinase inhibition. Our previous studies identified significant inhibition of allergic response in an acute HDM-induced model of asthma by selective pharmacological inhibition of AMCase [49], suggesting that AMCase is a contributor to an early inflammatory component in asthma. Interestingly, while AMCase is significantly induced in early stages of HDM administration [49], macrophage-specific CHIT1 localization in lungs is predominant at later stages and correlates with airway remodeling development. Importantly, immunohistochemical localization studies for the first time revealed CHIT1 involvement in human fatal asthma, where CHIT1 was localized in the surroundings of severely remodeled airways and AMCase was not present. Therefore, we claim that CHIT1 is a valid therapeutic target in tissue remodeling processes and fibrosis occurring in severe asthma.

Chitinases modulate TGFβ1 inflammatory and tissue remodeling activities [43,44,50,51] which correlate with asthma severity and fibrotic processes development. OATD-01 in a dose-dependent manner inhibited active TGFβ1 release, a macrophage-derived cytokine [52] and a known driver of subepithelial fibrosis in asthma. This finding is relevant to asthma patients, since TGFB1 mRNA is increased in bronchial biopsies from asthmatic individuals and its levels correlate with a degree of subepithelial fibrosis [53,54]. Crucial evidence from murine animal models shows that inhibition of TGF-β1 reduces allergen-induced subepithelial collagen deposition, and intratracheal instillation of TGF-β1 is sufficient to cause subepithelial fibrosis [55,56]. Our work suggests that inhibition of TGF-β1 production by profibrotic macrophages is regulated by CHIT1 and can explain how OATD-01 attenuates airway remodeling.

Overall, data presented here strengthen the rationale of the inhibition of profibrotic macrophages to reverse airway remodeling. We demonstrate that activation of CHIT1 is observed in fatal asthma, but more importantly we present OATD-01 as a safe and efficacious drug. Pharmacological inhibition for CHIT1 can be a treatment of various chronic lung diseases with ongoing tissue remodeling processes.

## 4. Materials and Methods

### 4.1. Human Sample Collection and Analysis

Lung tissues were obtained from 12 cases of fatal asthma and 9 control cases (subjects who died of non-pulmonary causes) who had their necropsies performed at the São Paulo Autopsy Service (SVOC) between 2002 and 2007. All individuals included in the asthmatic group had a history of asthma and died in an acute respiratory crisis. The deaths were ascribed to asthma by pathologists during post-mortem analysis. Chitinase immunostained slides were scanned using a Pannoramic Flash Scanner (3DHistech, Budapest, Hungary). Chitinase expression was assessed in 3 to 4 small airways randomly selected using Image-Pro Plus^®^ image-analysis software, version 6.0 for Windows^®^ (Media Cybernetics, Silver Spring, MD, USA). Small airways were defined as those showing a basement membrane (BM) perimeter ≤ 6 mm. The analyzed region comprised the area between the epithelial basement membrane to the adventitia layer. Chitinase-positively stained area was normalized by the corresponding airway basement membrane perimeter, results being expressed as μm^2^/μm. This study was approved by the Institutional Ethical Board, University of São Paulo, No. of approval 360/12.

### 4.2. Animal Studies

All in vivo studies were performed in accordance with the EU Directive 2010/63/EU and the Polish legislation for animal experiments of the Polish Ministry of Science and Higher Education (26 February 2015) and approved by the Local Ethics Committee for the Animal Experimentation in Warsaw, Poland (Approvals No. WAW2/65/2014, WAW2/73/2014, WAW2/11/2015, WAW2/79/2015, WAW2/213/2016).

### 4.3. HDM-Induced Chronic Airway Inflammation and Remodeling Mouse Model

Airway inflammation and remodeling was induced in 8-week-old female C57BL/6 mice (Charles River Laboratories, Sulzfeld, Germany) by intranasal administration of 40 μg of HDM extract (Greer Laboratories) 5 times a week for up to 7 weeks. OATD-01 was administered p.o. at 3 or 30 mg/kg in 20:80 solutol:5% glucose in distilled water (*v*/*v*) once a day for the indicated on figures time.

### 4.4. BAL Collection

Lungs were washed via trachea using 1 mL of PBS. The collected BAL fluid was centrifuged (10 min, 2000 rpm, 4 °C). Supernatant was collected for further analysis and cell pellet was resuspended in 300 μL of PBS and subsequently used for flow cytometry analysis.

### 4.5. Plasma Collection

Blood was collected from the vena hepaticae and centrifuged 10 min, 2000× *g* at room temperature. Plasma was collected and then frozen in −80 °C for subsequent analyses.

### 4.6. Chitinolytic Activity in Murine BALF and Plasma

The enzymatic activity of chitinases in murine BALF and plasma was measured as described previously [33].

### 4.7. Lung Collection

Lungs were washed with PBS via right ventricle post cutting abdominal aorta (as described [33]). Shortly, left lung was isolated and divided for further analyses. Right lung was prepared as described in “Histology” Section 4.12.

### 4.8. Flow Cytometry Analysis

The inflammatory cell influx was analyzed in the BAL fluid. Total BAL cells were counted with flow cytometry (Guava, Merck Millipore, Darmstadt, Germany). The inflammatory cell subpopulations (macrophages and eosinophils) were further characterized with flow cytometry using the relevant antibodies (Table 1).

### 4.9. Real-Time PCR

Gene expression analysis in murine lungs using real-time PCR method was performed as described previously [29].

### 4.10. ELISA

For quantification of active TGFβ1 level in BAL fluid ELISA (cat. No. 437707, eBioscience, San Diego, CA, USA) was used according to manufacturer’s protocol.

### 4.11. Western Blot Analysis

For Western blot analysis, BAL fluid samples (an equal volume of each sample) were resolved with SDS-PAGE, transferred to nitrocellulose membrane, and proteins were detected with appropriate primary (anti-mCHIT1, Cat No. AF5325, R&D Systems, Minneapolis, MN, USA) and secondary antibodies according to manufacturer’s protocols. The densitometry analysis results were presented as mean for each experimental group—Control, HDM, and OATD-01 (30 mg/kg) and are referred to HDM group as 1.

### 4.12. Histology

The collected lungs were prepared and processed as described previously [32]. The 5 µm lung sections were stained with hematoxylin and eosin (HE), Masson’s trichrome, periodic acid–Schiff stain (PAS), picrosirius red (PSR)–fast green (FG). or immunohistochemistry. After staining, slides were dehydrated, cleared, and mounted with synthetic resin-based medium. Analysis was performed under light microscope (PrimoStar, Zeiss, Jena, Germany) equipped with digital camera (Axiocam ERc5s, Zeiss).

### 4.13. Immunohistochemistry

Immunohistochemical staining of CHIT1 and AMCAse in murine lung tissue slides was performed as previously described [28].

### 4.14. Semi-Quantitative Scoring of Goblet Cell Metaplasia in HDM-Induced Mouse Model of Asthma

PAS-stained sections were assessed by experienced pathologist, using published scoring systems (scale 0–4), to evaluate blindly goblet cell metaplasia [57]. From every animal, 3 sections separated by 200 µm were prepared, and from every section 3–5 bronchioles of similar size (perimeter 500 ± 200 µm) were taken for analysis. The average score per animal was calculated and used for statistical analysis.

### 4.15. Quantitative Analysis of Bronchiole Collagen Wall Thickness and Bronchiole Epithelium Thickness in HDM-Induced Mouse Model of Asthma

Quantitative analyses were performed using ImageJ software on sections stained with Masson’s trichrome. From every animal, 3 sections separated by 200 µm were prepared, from them, a total of 30 bronchioles with perimeters of less than 1000 µm were chosen, photographed, and analyzed blindly. Thickness was measured by manual circumventing of blue area (collagen) around bronchiole and red area of epithelium of bronchiole, as well as the length of basal membrane. Results were calculated and presented as area/µm of basal membrane and used in statistical analyses.

### 4.16. OATD-01 Pharmacokinetics/Pharmacodynamics (PK/PD) Studies in Mice

We evaluated PK/PD profile of OATD-01 in a single dose or 2-week-long dosing (30 mg/kg, PO qd) in mice subjected to HDM treatment. Further analysis of PK/PD properties of compound OATD-01 in plasma and lung tissue homogenates was performed as described previously [30].

### 4.17. Statistical Analysis

Data were analyzed using GraphPad Prism v. 7.0. Outliers were identified by ROUT method with 1% threshold. Parametric one-way ANOVA with Dunnett’s test for multiple comparisons was used to evaluate the differences between the groups with Gaussian distribution of results (verified with D’Agostino and Pearson omnibus normality test and Shapiro–Wilk normality test). In case of non-Gaussian distribution, non-parametric Kruskal–Wallis test with Dunn’s multiple comparison test was used to evaluate the differences between the groups to determine *p* values. *p*-values < 0.05 were considered significant and noted with asterisks (* for *p* < 0.05, ** for *p* < 0.01, *** for *p* < 0.001, **** for *p* < 0.0001).

## 5. Conclusions

OATD-01 exhibited anti-inflammatory efficacy in a chronic HDM-induced asthma model by reducing CD45 cells in BALF that correlated with decreased chitinolytic activity in BALF and serum. More importantly, OATD-01 attenuated airway remodeling in HDM model by inhibition of CHIT1-mediated active TGFβ1 release and subsequent Th2-dependent IL-13 production. Administration of OATD-01 in a therapeutic regimen revealed the importance of the CHIT1-dependent macrophage–fibroblast positive feedback loop. The overall effect of CHIT1 inhibition resulted in less collagen deposition in extracellular space and attenuated airway remodeling observed in severe asthma. Our conclusions and observations extend beyond severe asthma, presenting CHIT1 inhibition as a novel and attractive therapeutic target for other disorders regulated by profibrotic macrophages, specifically IPF and other ILDs.

## Figures and Tables

**Figure 1 ijms-24-04719-f001:**
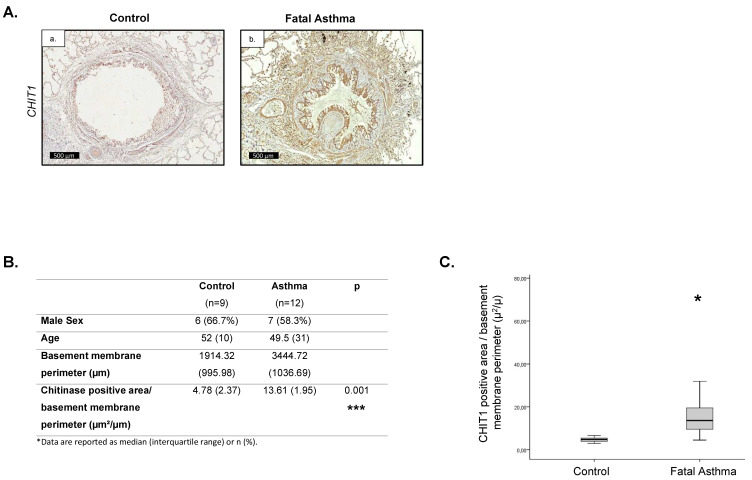
Increased contribution of CHIT1 in airway remodeling in fatal asthma patients. (**A**) Representative immunohistochemical staining of CHIT1. (**B**) Table containing donor and fatal asthmatics characteristics, basement membrane perimeter lengths, and CHIT1-positive area/basement membrane perimeter ratios (µ^2^/µ). (**C**) Quantitative representation of CHIT1-positive area/basement membrane perimeter ratios (µ^2^/µ) in the lungs of donors and fatal asthma individuals. Data presented as mean ± s.e.m. *p*-values < 0.05 were considered as statistically significant and presented as * for *p* < 0.05 and*** for p < 0.001.

**Figure 2 ijms-24-04719-f002:**
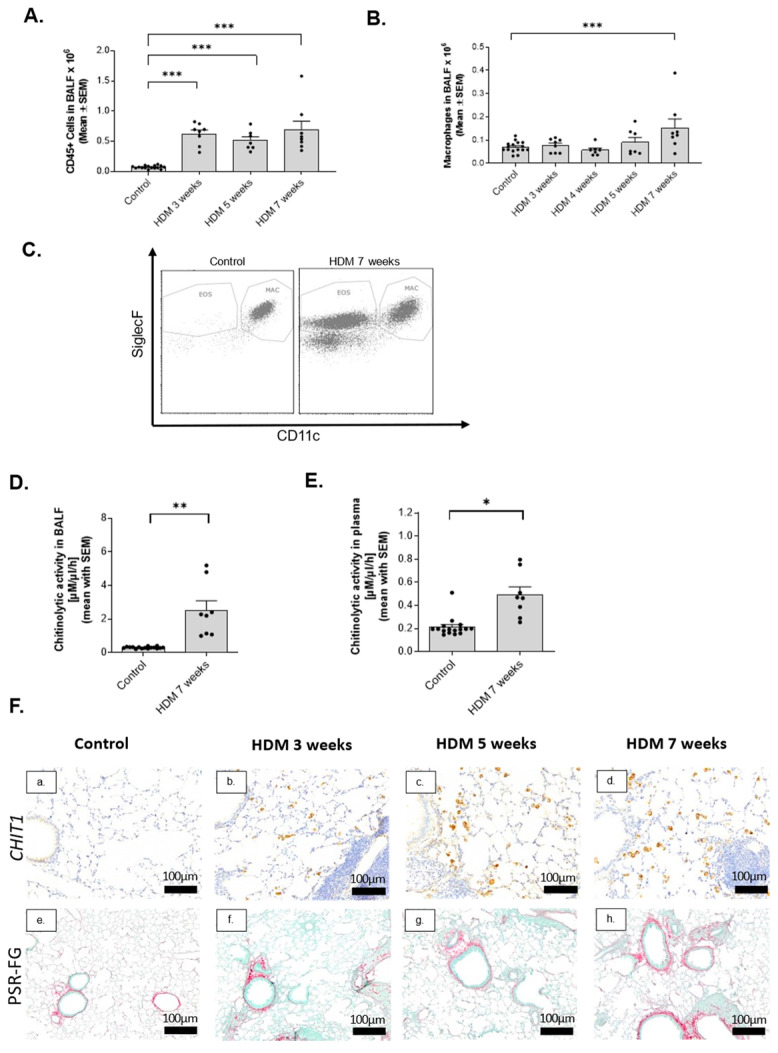
Activation of chitinases in chronic HDM-induced airway remodeling model in mice. (**A**) Number of total infiltrated leukocytes (CD45-positive cells) in BAL fluid analyzed by flow cytometry during progression of chronic airway remodeling at 3–7 weeks of HDM administration intranasally 5 times a week. (**B**) Flow cytometry analysis of alveolar macrophages (Siglec F+, CD11c+) represented as total macrophage numbers in BAL fluid (1 mL) tested after 3, 4, 5, and 7 weeks of HDM administration. (**C**) A representative dot plot of BALF-isolated CD45-positive cells (Siglec F+, CD11c+) representing alveolar macrophages. (**D**) Chitinolytic activity in BAL fluid and (**E**) plasma recovered from naïve mice challenged with PBS or HDM (7 weeks of administration). (**F**,**a**–**h**) Representative histological pictures of samples immunostained for CHIT1 (**a**–**d**) and picrosirius red–fast green (**e**–**h**) for collagen on mouse lungs; HDM-induced airway remodeling progression was followed at timepoints 3 weeks (**b**,**f**), 5 weeks (**c**,**g**), and 7 weeks (**d**,**h**) of intranasal administration of HDM. Control samples are presented on a and e panels. Data presented as mean ± s.e.m. *p*-values < 0.05 were considered as statistically significant and presented as * for *p* < 0.05, ** for *p* < 0.01, *** for *p* < 0.001.

**Figure 3 ijms-24-04719-f003:**
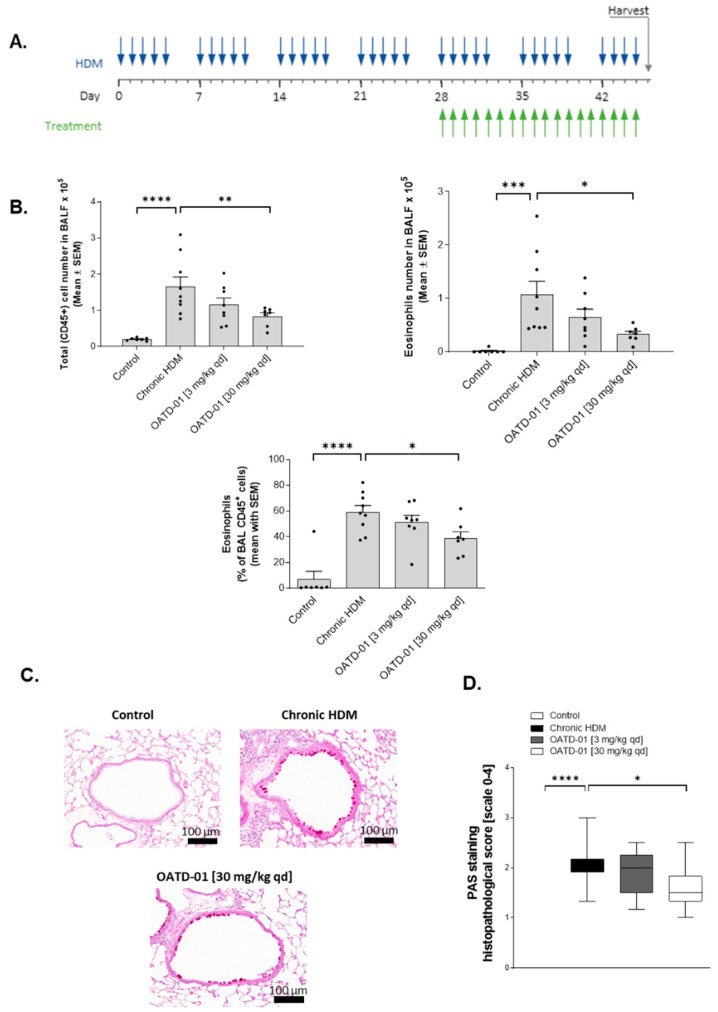
OATD-01 reduces goblet cell metaplasia and exhibits anti-inflammatory effects in chronic HDM-induced model. (**A**) Schematic representation of HDM model and OATD-01 administration regimen. Blue arrows indicated HDM installations while green OATD-01 treatment. (**B**) Flow cytometry analysis of pulmonary inflammation as represented by total leukocyte numbers (CD45+ cells) and eosinophil population (Siglec F+, CD11c- cells) in particular, in BAL fluid of mice subjected to chronic HDM (7 weeks) and treated with OATD-01 (3 and 30 mg/kg; PO; qd) or vehicle controls. (**C**) Representative PAS staining of goblet cell metaplasia in chronic HDM-induced airway remodeling model. (**D**) Semi-quantitative analysis of goblet cell metaplasia by scoring system (0–4) in the lung sections of chronic HDM administration model as compared to vehicle treated controls. Data presented as mean ± s.e.m. *p*-values < 0.05 were considered as statistically significant and presented as * for *p* < 0.05, ** for *p* < 0.01, *** for p < 0.001, and **** for *p* < 0.0001.

**Figure 4 ijms-24-04719-f004:**
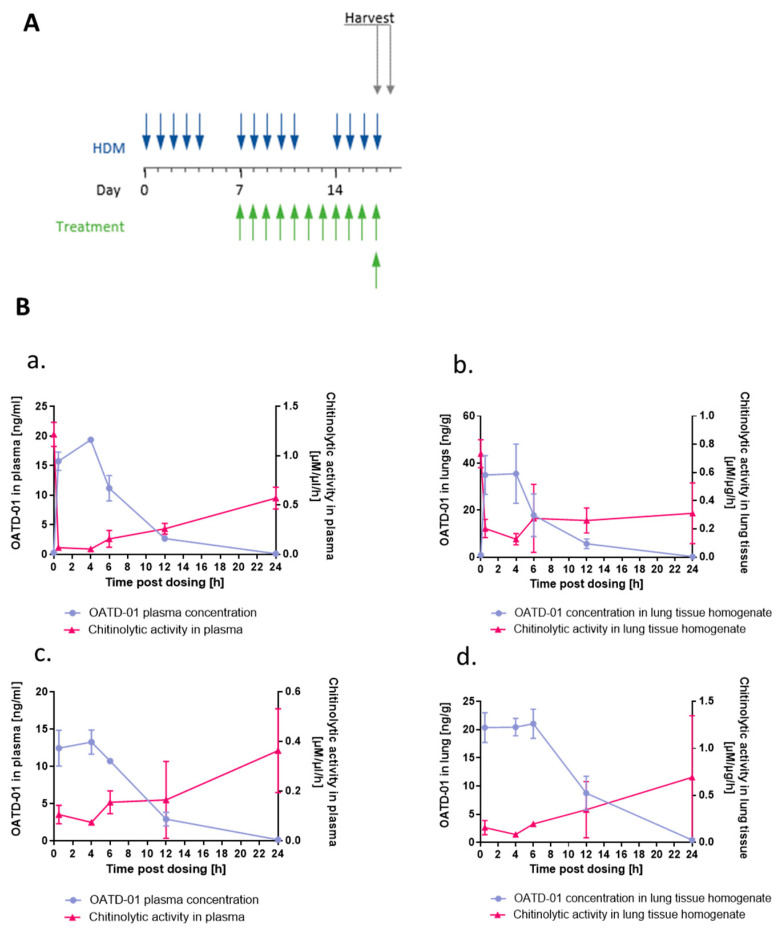
OATD-01 decreases chitinolytic activity in BAL fluid and plasma in a dose-dependent manner. (**A**) Schematic representation of PK/PD study in 3-week-long HDM model with two OATD-01 administration regimens (single dose at 30 mg/kg or 14 day). Blue arrows indicated HDM installations while green OATD-01 treatment. (**B**) Graphs showing relation between chitinolytic activity (magenta) and OATD-01 concentration (blue) in plasma (**a**,**c**) and lung homogenates (**b**,**d**) following single dose of OATD-01 at 30 mg/kg on the last day of the study (**c**,**d**) or multiple administrations of OATD-01 at 30 mg/kg once a day (**a**,**b**).

**Figure 5 ijms-24-04719-f005:**
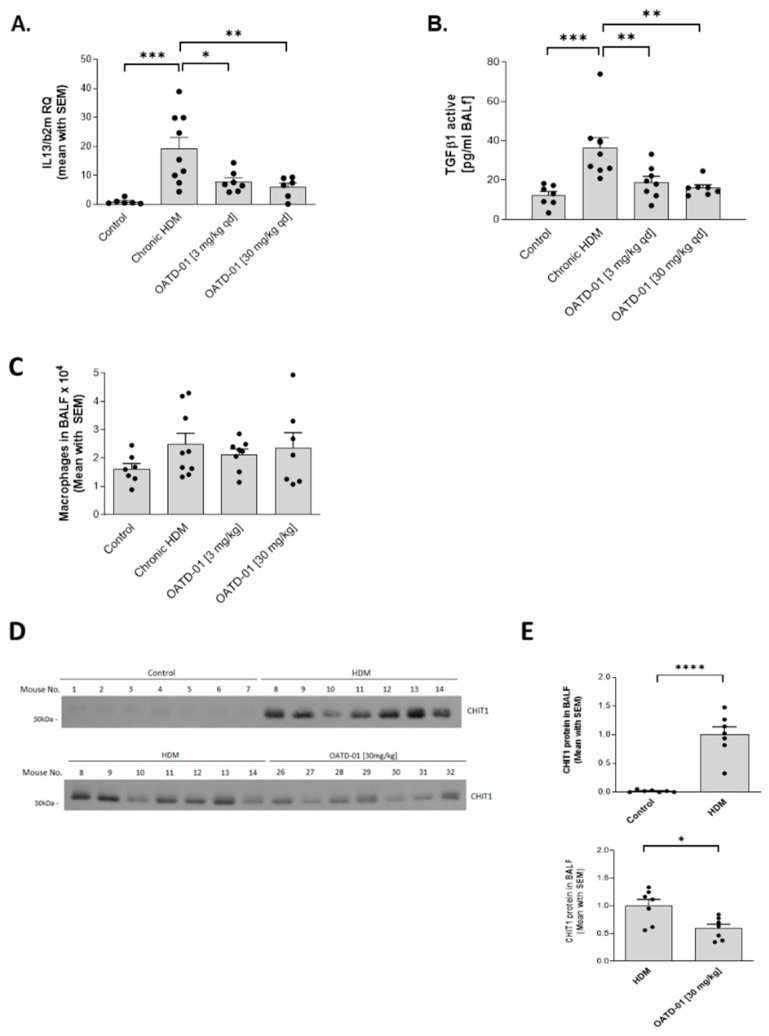
OATD-01 ameliorates TGFβ1, IL-13, and CHIT1 in chronic HDM model in a dose-dependent fashion. (**A**) mRNA expression of Il13 in the lungs and (**B**) active TGFβ1 levels in BAL fluid in mice with chronic HDM (7 week) treated with OATD-01 (3 and 30 mg/kg; PO; single dose) or vehicle control. (**C**) Number of macrophages in BAL fluid in mice with chronic HDM (7-week-long) treated with OATD-01 (3 and 30 mg/kg; PO; single dose) or vehicle control. The level of CHIT1 protein in BAL fluid of control and HDM-instilled mice treated with vehicle or OATD-01 (30 mg/kg) in chronic airway inflammation model as evaluated by (**D**) Western blot and (**E**) corresponding densitometry analysis. Data presented as mean ± s.e.m. *p*-values < 0.05 were considered as statistically significant and presented as * for *p* < 0.05, ** for *p* < 0.01, *** for *p* < 0.001, and **** for *p* < 0.0001.

**Figure 6 ijms-24-04719-f006:**
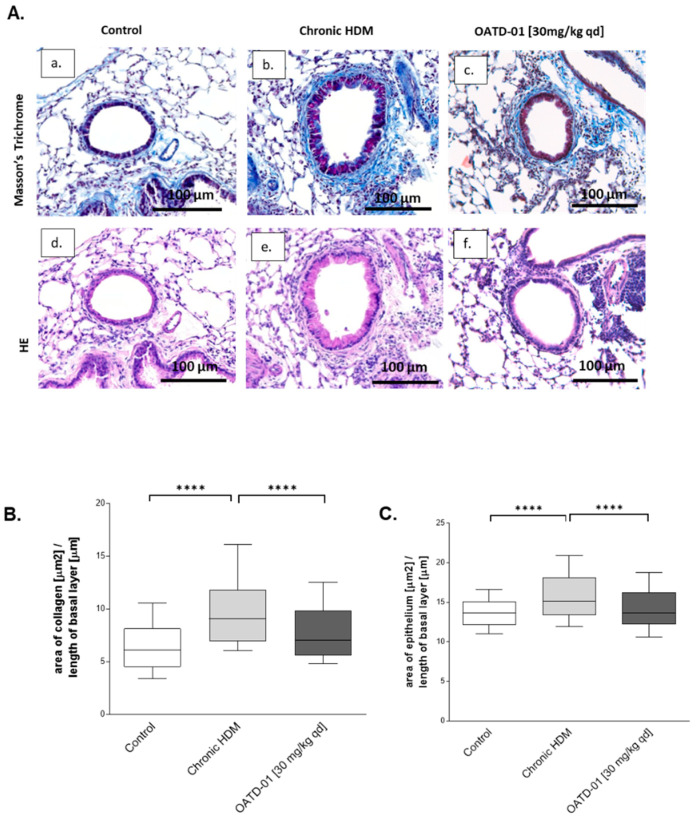
OATD-01 reduces features of airway remodeling after chronic HDM administration. (**A**) Histological evaluation of airway remodeling (bronchiole collagen wall and epithelial thickness) as represented by Masson’s trichrome (**a**–**c**) and hematoxylin and eosin (**d**–**f**) in a 7-week-long chronic HDM model in C57BL/6 mice treated with OATD-01 (30 mg/kg; PO; qd) (**c**,**f**) or vehicle control; Control (**a**,**d**) and Chronic HDM (**b**,**e**). (**B**) Quantitative analysis of peribronchial collagen area (collagen wall thickness) stained with Masson’s trichrome in a chronic HDM model with or without OATD-01 treatment (30 mg/kg; PO; qd; n = 8). (**C**) Quantitative analysis of bronchiole epithelial thickness, measured at the same time as analysis of collagen wall thickness (on Masson’s trichrome-stained samples) in a HDM model with or without OATD-01 treatment (30 mg/kg; PO; qd; n = 8). Data presented as mean ± s.e.m. *p*-values < 0.05 were considered as statistically significant and presented as **** for *p* < 0.0001.

**Figure 7 ijms-24-04719-f007:**
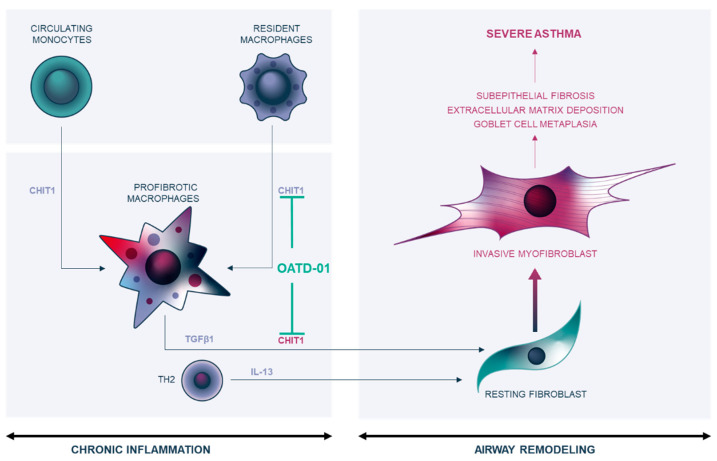
Inhibition of macrophage-specific CHIT1 attenuates airway remodeling. Decreased macrophage-specific CHIT1 expression and activity changes phenotype of profibrotic macrophages and blocks TGFβ1 and subsequent Th2-dependent IL-13 production. OATD-01 inhibits CHIT1-dependent macrophage–fibroblast positive feedback loop leading to fibroblast activation and myofibroblast transition. The overall effect of CHIT1 inhibition results in less collagen deposition in extracellular space, decreased goblet cell hyperplasia, and attenuated airway remodeling—features associated with severe asthma.

**Table 1 ijms-24-04719-t001:** Antibodies used for flow cytometry analysis.

Antigen	Conjugate	Dilution	Cat.	No.
Gr1	Alexa 488	1:800	Biolegend, San Diego, CA, USA	108417
Siglec F	PE	1:400	BD Biosciences, San Jose, CA, USA	552126
CD11c	APC	1:400	eBiosciences San Diego, CA, USA	17-0114-82
CD45.2	PerCp-Cy5	1:100	BD Biosciences, San Jose, CA, USA	552950

## Data Availability

The data presented in this study are available upon request.

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
