# Peer review of "Inhibition of Macrophage-Specific CHIT1 as an Approach to Treat Airway Remodeling in Severe Asthma"

_ijms, 2023, doi:10.3390/ijms24054719_

Round 1

Reviewer 1 Report

This is an interesting article by Sklepkiewicz et al. I would like to address a small number of suggestions to authors that may improve the manuscript.

General recommendations

Please check all the text for spelling mistakes.

Please correct all references according to journal stile.

References should be described as follows:

Journal Articles:

1.      Author 1; Author 2. Title of the article. Abbreviated Journal Name Year; Volume: page range.

Results

The CHIT1 is produced and secreted both by macrophages and neutrophils. Moreover, epithelial cells can also secrete CHIT1.

In this study, authors do not used neutrophils' markers to show that neutrophils had absent or extremely low level.

Figure 2 In this figure authors report cell’s types in BAL fluid analyzed by flow cytometry. Unfortunately, a flow cytometry diagram was not presented. Please add to figure 2

Page 5, line 138 please write in vivo in italics 

Please write analytically which kit ELISA was used for quantification of active TGFβ1 level in BAL fluid. Add country and city for eBioscience.

Conclusions are missing.

Author Response

Reviewer 1

This is an interesting article by Sklepkiewicz et al. I would like to address a small number of suggestions to authors that may improve the manuscript.

We are thankful to the reviewer for spending time to improve our manuscript and appreciate interest in significance of our findings. Changes in the manuscript are marked in red.

General recommendations

Please check all the text for spelling mistakes.

We have checked all the text for spelling and grammar mistakes.

Please correct all references according to journal stile. References should be described as follows:

Journal Articles:

  1. Author 1; Author 2. Title of the article. Abbreviated Journal Name Year;Volume: page range.

In the revised version of the manuscript references are in the format required by journal guidelines.

The CHIT1 is produced and secreted both by macrophages and neutrophils. Moreover, epithelial cells can also secrete CHIT1.

In the revised version of the manuscript we have acknowledge that CHIT1 can be expressed by macrophages, neutrophils and epithelial cells (lines 46-48), however our own data strongly suggests that in the model we used and human subjects we examined this is predominantly macrophages-dependent expression.

In this study, authors do not used neutrophils' markers to show that neutrophils had absent or extremely low level.

Indeed we have used GR.1 antibody that does not allow for discrimination between inflammatory monocytes and neutrophils (Ly6C and Ly6G respectively), however this population was small and has not changed with treatment of our compound therefore we have not investigated it further. However after reviewer’s comment we have changed conclusions and commented about a population of GR.1 positive cells representing neutrophils and inflammatory monocytes (lines 96 to 99).

Figure 2 In this figure authors report cell’s types in BAL fluid analyzed by flow cytometry. Unfortunately, a flow cytometry diagram was not presented. Please add to figure 2

We have added a representative dot-plot gated on alveolar macrophage population to present gating of our cell of interest (New Figure 2C, lines 114-115).

Page 5, line 138 please write in vivo in italics 

We have corrected this typo.

Please write analytically which kit ELISA was used for quantification of active TGFβ1 level in BAL fluid. Add country and city for eBioscience.

We have added this information in the revised version of the manuscript.

Conclusions are missing.

We have added a conclusion section (lines 192-203) to summarize major findings of our work.

Reviewer 2 Report

The study of the mechanisms of asthma is of great clinical interest because of its high prevalence and great social importance.

Comments:

1. It is recommended that the abstract be structured to be more reader-friendly.

2. It is recommended to add a figure with the study design

3. It is recommended to add more clinical data on patients with asthma and the control group. What were the inclusion criteria for each of the groups? Did these patients smoke, have other conditions such as COPD?

4. It is recommended that a conclusion section be added to summarize the findings and future research perspectives.

Author Response

Reviewer 2

The study of the mechanisms of asthma is of great clinical interest because of its high prevalence and great social importance.

We are thankful to the reviewer for spending time to improve our manuscript and greatly appreciate noticing clinical significance of our findings. Changes in the manuscript are marked in red.

Comments:

  1. It is recommended that the abstract be structured to be more reader-friendly.

We have followed guidelines for abstract preparations which is requiring one body text without sections.

  1. It is recommended to add a figure with the study design

We have presented our animal study design in Fig 3A and Fig.4A – we will make sure they are big enough to be visible.

  1. It is recommended to add more clinical data on patients with asthma and the control group. What were the inclusion criteria for each of the groups? Did these patients smoke, have other conditions such as COPD?

Conditions such as smoking were not controlled in individuals in the study, however their primary diagnosis by physician was severe asthma and not COPD. We have added a sentence in the section describing human material to address inclusion criteria: All individuals included in the asthmatic group had a history of asthma and died in an acute respiratory crisis. The deaths were ascribed to asthma by pathologists during post-mortem analysis (lines 301-303).

  1. It is recommended that a conclusion section be added to summarize the findings and future research perspectives.

We have added a conclusion section (lines 192-203) to summarize major findings of our work.